# Preoperative Computer-Assisted Laparoscopy Planning for the Minimally Invasive Surgical Repair of Hiatal Hernia

**DOI:** 10.3390/diagnostics10090621

**Published:** 2020-08-21

**Authors:** Silviu Daniel Preda, Cătălin Ciobîrcă, Gabriel Gruionu, Andreea Șoimu Iacob, Konstantinos Sapalidis, Lucian Gheorghe Gruionu, Ștefan Castravete, Ștefan Pătrașcu, Valeriu Șurlin

**Affiliations:** 1Department of Surgery, University of Medicine and Pharmacy of Craiova, 200349 Craiova, Romania; sdpreda@gmail.com (S.D.P.); vsurlin@gmail.com (V.Ș.); 2Faculty of Automation, Computers and Electronics, University of Craiova, 200776 Craiova, Romania; catalin.ciobirca@yahoo.com (C.C.); andr33a_soimu@yahoo.com (A.Ș.I.); 3Faculty of Mechanics, University of Craiova, 200512 Craiova, Romania; gruionu@gmail.com (G.G.); lgruionu@gmail.com (L.G.G.); 4Restore Surgical LLC, Arlington, MA 02474, USA; 53rd Department of Surgery, “AHEPA” University Hospital, Aristotle University of Thessaloniki, Medical School, 54621 Thessaloniki, Greece; sapalidiskonstantinos@gmail.com; 6Caelynx Europe Ltd., 200374 Craiova, Romania; scastravete@caelynx.ro

**Keywords:** laparoscopy, surgical planning, training, hiatal hernia

## Abstract

Minimal invasive surgical procedures such as laparoscopy are preferred over open surgery due to faster postoperative recovery, less trauma and inflammatory response, and less scarring. Laparoscopic repairs of hiatal hernias require pre-procedure planning to ensure appropriate exposure and positioning of the surgical ports for triangulation, ergonomics, instrument length and operational angles to avoid the fulcrum effect of the long and rigid instruments. We developed a novel surgical planning and navigation software, iMTECH to determine the optimal location of the skin incision and surgical instrument placement depth and angles during laparoscopic surgery. We tested the software on five cases of human hiatal hernia to assess the feasibility of the stereotactic reconstruction of anatomy and surgical planning. A whole-body CT investigation was performed for each patient, and abdominal 3D virtual models were reconstructed from the CT scans. The optical trocar access point was placed on the xipho-umbilical line. The distance on the skin between the insertion point of the optical trocar and the xiphoid process was 159.6, 155.7, 143.1, 158.3, and 149.1 mm, respectively, at a 40° elevation angle. Following the pre-procedure planning, all patients underwent successful surgical laparoscopic procedures. The user feedback was that planning software significantly improved the ergonomics, was easy to use, and particularly useful in obese patients with large hiatal defects where the insertion points could not be placed in the traditional positions. Future studies will assess the benefits of the planning system over the conventional, empirical trocar positioning method in more patients with other surgical challenges.

## 1. Introduction

Minimally invasive surgery such as laparoscopy allows faster postoperative recovery with less trauma, inflammatory response, and cosmetic scarring [1]. Although operator-dependent, laparoscopy requires standardization and planning. In particular, hiatal hernias repair via laparoscopy requires access to a deep and narrow region and a higher degree of expertise of the surgeon. Moreover, it involves a good exposure and positioning of the surgical ports in order to ensure good triangulation of the surgical instruments, optimal ergonomics, correct distance to the targeted area, and provide better angles on the body surface and between instruments to avoid the fulcrum effect of the long and rigid instruments.

Although the principles of trocar positioning are well established, the strategy may vary according to a multitude of factors, such as anatomic considerations, pathological particularities, technical limitations, or esthetic reasons. In today’s practice, as there is no virtual planning method widely adopted, it is the role of the surgeon to establish a given intraoperative strategy, which is based on his/her experience and the surgical dogma [2]. The heuristic model is inferior in terms of ergonomics, rehearsal tool, and educational support, while it is does not offer a detailed evaluation of the operative field [3].

In the last twenty years, several approaches to optimize the trocar positioning in laparoscopic surgery have been pursued [4,5,6,7]. In one of the earliest studies dealing with the operative planning, Hanna et al. emphasized the role of trocars placement and working angles of the laparoscopic instruments on endoscopic manipulation [4]. Simple parameters like execution time and performance quality assessment were analyzed in order to obtain optimal port location for laparoscopic suturing.

Contrast-enhanced computed tomography scan of patients with large hiatal hernias are performed more often, replacing the traditional barium contrast x-ray, because it offers more information about the relationships with surrounding structures, allows the measurement of the hiatal orifice, and is particularly useful for reconstruction.

During the last two decades, computer software for 3D reconstruction has allowed the patient’s anatomy to be reconstructed accurately, as well as provide a stereotactic orientation of the surgeon [8,9,10]. This is useful to personalize the surgical approach by adapting to the variable anatomy and pathologic morphology of each patient. Unfortunately, the simulation platforms for laparoscopy only use a generic mannequin not a realistic 3D anatomy for learning the laparoscopy techniques.

To address this issue, in the present study, we developed a novel surgical planning software, iMTECH, which not only generates a virtual 3D model of the patient’s anatomy but also allows simulation of the surgical tool placement for optimal trocar port positioning. We tested the feasibility of a surgical planning software on patient’s CT scans laparoscopic approach for hiatal hernia repair. We chose hiatal hernia surgery due to the fact that the intraoperative position of the hiatus consistently matches that of the tomography, with negligible errors for the diaphragmatic movement during breathing and pneumoperitoneum. This laparoscopy module allows for preoperative planning for increased ergonomics in demanding laparoscopic surgery (i.e., knot tying, suturing, tissue manipulation).

## 2. Materials and Methods

### 2.1. Human Subjects

Five patients (2 males and 3 females, aged 46–65 years) who were scheduled for a CT exam and laparoscopic surgery were included in the pre-surgery planning study. The patient’s body mass indices were 36.5, 24.5, 37.6, 25.3 and 28.9 kg/m^2^. Informed consent was obtained before the procedure for all five patients. The study protocol was approved by the Research and Ethics Committee of the University of Medicine and Pharmacy of Craiova (No. 68/23.02.2017) and carried out in accordance with the Code of Ethics of the World Medical Association (Declaration of Helsinki) for experiments involving humans.

The CT scans were performed with intravenous contrast (three patients) and oral contrast (two patients). CT images were acquired using a 20-slice Siemens Biograph mCT equipment (Siemens Healthcare, Minchen, Germany), with a 512 × 512 reconstruction matrix. A chest, abdominal and pelvic CT investigation was performed for each patient, including a non-contrast examination followed by a post intravenous iodine contrast (Iohexol-Omnipaque™ 350 mg L/mL) examination. The acquisition was performed at 5 mm intervals.

### 2.2. Three-Dimensional Reconstruction of the Patient’s Anatomy

We developed the laparoscopy planning module as part of our existing medical imaging software platform, iMTECH [11]. The medical imaging software generates a 3D volumetric reconstruction of the patient’s anatomy from the series of DICOM files generated during the CT scan: 526 slices at 1 mm slice interval (patient 1), 337 slices at 1.3 mm interval (patient 2), 442 slices at 1.5 mm interval (patient 3), 544 slices at 1 mm slice interval (patient 4) and 128 slices at 2.5 mm slice interval (patient 5). The 3D models of the five patients had a total number of 115.868 million voxels, 88.343 million voxels, 137.888 million voxels, 142.606 million voxels, and 33.554 million voxels, respectively. The value can be adjusted to a lower resolution of 10 million voxels or a medium resolution of 50 million voxels.

The 3D representation can be adjusted to show different anatomical structures, such as air, bone, large blood vessels, skin, adipose tissue, etc. The laparoscopy planning module allows the insertion and manipulation of forceps and camera in the 3D rendered representation of the patient, allowing representation of the instrument’s tip on the 2D MPR axes. The software calculates the trocar’s position: distance between trocars, distance from the site on the skin for trocar insertion to the desired area, and the angles between the forceps and camera.

### 2.3. Laparoscopy Planning Module

The laparoscopy module contains the following virtual tools: forceps, which is a 3D rendered representation of a laparoscopic forceps; camera, with similar graphic features as the forceps; basic tool which is a configurable cylindrical 3D representation with adjustable length and radius; and references, represented as small spheres. Another element is the distance function, which allows the calculation of the physical distance between different tools.

The forceps tool is a 3D representation of a forceps superimposed on the 3D rendered patient’s anatomy. The tip of the forceps can be precisely placed on an anatomical target in the axial, sagittal and coronal projections of the CT scan. After identifying the desired work area (the hiatus) and fixing the tip of the forceps on the surgical target, the operator can move the forceps by dragging the handle with the computer cursor. The intersection of the forceps with the skin of the patient is marked with a reference sphere on the forceps’ shaft. The distance from the reference sphere to the tip of the forceps is the shortest vertical path from the surgical target to the skin and is displayed on the left panel of the software graphical interface. After placing the second forceps, the angle between the two forceps is automatically calculated and displayed.

The planning software calculates the curvilinear “skin” distance, D(s), between the skin access points of any two forceps or camera tools, and the real distance, D(d), between the tip of the instruments at the level of the peritoneum.

However, for a more practical assessment of the positioning of the working trocars and an easier translation to the clinical setting, a set of coordinates has been implemented based on a geometric triangular model (Figure 1). The access point (P) is defined by two values–corresponding to specific distances–and the 90° angle between the two vectors, conformable to a basic two-dimensional Euclidian model:x1 = the distance on a perpendicular vector between the median line and the access pointy1 = the distance on the median line between the xiphoid and the intersection of the sagittal plane with the xipho-umbilical line.

Moreover, the simulation software allows calculation of the optimal distance between trocars based on the following determinants: the azimuth angle (α_a_), the manipulation angle (α_m_), and the elevation angle (α_e_). The azimuth angle is the angle created by the longitudinal axis of any of the active instruments and the endoscopic camera. The manipulation angle is defined as the angle created by the two active instruments. The elevation angle of the instrument is formed by the axis of the active instrument and the horizontal plane.

### 2.4. Laparoscopic Procedure

The laparoscopic equipment consisted of conventional 0° and 30° camera (Karl Storz, Tuttlingen, Germany), pyramidal tip trocars, conventional laparoscopic instruments (such as scissors, needle holders, and graspers), monopolar hook, and advanced bipolar cauterization unit (LigaSure™, Medtronic, Minneapolis, MN, USA). All procedures were recorded on a hard disk and analyzed postoperatively by the members of the surgical team. The intraoperative data (incidents, blood loss, duration of the procedure, etc.) were also recorded.

### 2.5. Control Group

Results were compared with a retrospective control group of patients who underwent laparoscopic repair of hiatal hernia. The control group consisted of 10 patients (all females), aged between 41 and 74 years, who were admitted and treated in our Department between February 2017 and March 2019. Only large type hiatal hernias were included in the control group. Patient charts, operating records, and surgical footage were analyzed, and essential data was recorded. All cases, both pre procedure planning group and control group were operated by the same surgical team.

We compared overall operating time and specific duration of essential surgical tasks: dissection (dissection of the hernia sac and of the hiatus, mobilization of the greater curvature of the stomach), cruroraphy, and creation of fundoplication.

## 3. Results

### 3.1. Surgical Planning Procedure

The planning procedure consists of determining the distances and angles between the abdominal access points and a given anatomical landmark (e.g., pubis, xiphoid) or an optical tracker placed on the surface of the abdomen, which allowed the operator to optimally position the trocars (Figure 2).

All the initial coordinates for the access points [P(x_n_y_n_)] were selected by the operators based on the conventional training for laparoscopic surgery. Given the initial coordinates and constraint to maintain the optimal angles α_m_ = 60°, and α_a_ = 30°, the software adjusted the coordinates of the access points for the four working trocars for each of the five virtual models corresponding to each patient (Table 1).

The optical trocar access point [P(opt)] was placed on the xipho-umbilical line and therefore was defined only by the distance to the xiphoid. The skin distance between the insertion point of the optical trocar and the xiphoid was 159.6, 155.7, 143.1, 158.3, and 149.1 mm, respectively, for an elevation angle of 40° (Figure 3). The position of trocars could be further adjusted manually according to the other intraoperative requirements or the preference of the surgeons.

### 3.2. Validation in the Clinical Setting

All patients successfully underwent the laparoscopic procedures. The procedures included five large hernia repairs using the floppy Nissen fundoplication performed by an experienced team of laparoscopic surgeons. The insufflation pressure usually varied between 9 and 11 mmHg. Five trocars were used for each intervention: one 10 mm optical trocar, two 10 mm working trocars, and two 5 mm working trocars. The mean operative time for the preoperative planning group was 170 min and for the control group 184.5 min. Operative time was further stratified into duration for essential surgical steps: dissection time (dissection of pars flaccida and pars condensa, dissection of hernia sac and esophageal hiatus, dissection of intrathoracic esophagus and mobilization of greater curvature of the stomach through dissection of the gastrosplenic ligament), cruroraphy time, and fundoplication time. For the preoperative planning group, mean dissection time was 43.6 min compared to 48.5 min for the control group. Similar decreases in duration were observed both in the duration of cruroraphy (20.8 min vs. 27.5 min) and in duration of fundoplication (29.2 min vs. 34.8 min). There were no cases of bowel or vascular injury. The clinical and operative particularities of each patient are presented in Table 2.

The four members of the team independently tested and evaluated the preoperative planning software. Postoperatively, a qualitative feedback survey based on an eight-point mixed Likert scale and open answer questionnaire assessed the surgical team’s response on issues such as the ease of use, complexity, and utility of the planning system. The general feedback was that iMTECH significantly improved the ergonomics, was easy to use, and was particularly useful in obese patients with large hiatal defects.

## 4. Discussion

Although laparoscopic surgery became the gold standard of surgery for complex cases of hiatal hernia, the variability of the patient’s BMI, anatomic, topometric and pathological changes requires a customized placement of the surgical ports. There are already surgical simulators and simulation software available on the market, but they use a generic anatomy and placement of the trocars, and do not allow extensive customization of the procedure for the patient’s anatomy [12,13,14,15]。

To address this problem, our laparoscopy module allows preoperative planning for increased ergonomics in laparoscopic demanding surgery (i.e., knot tying, suturing, tissue manipulation) through a better placement of access ports and better orientation of instruments inside the abdomen based on the patient’s anatomy. Moreover, it avoids any interference and the fulcrum effect between rigid surgical instruments.

The software is especially useful when the operative field is in narrow surgical spaces, such as the pelvic area or the gastro-esophageal junction. The difficulty of the laparoscopic approach lies in the anatomical and pathophysiologic particularities of these regions, as well as the sharp angles of approach between the instruments, which makes the correct placement of trocars crucial for good intraoperative ergonomics. Moreover, a detailed understanding of the local topography and relation with the adjacent structures is mandatory for good intra- and postoperative outcomes.

In the present study, the optimization of the geometric and kinematic requirements was directed by the operator. However, if needed, the software can provide a model of optimum preoperative planning starting from predetermined values for the distance and angle of each instrument. Moreover, the software can track the trajectory of the instruments in the limited space of the subdiaphragmatic region in order to verify the limitation of movement imposed by the nearby solid anatomical structures.

As part of laparoscopic procedure, CO_2_ is insufflated into the abdomen to create a working space for surgical instruments, a pneumoperitoneum. Although available in the software package, for these particular surgical procedures, the pneumoperitoneum was not simulated in the present study as changes in the intraoperative parameters in the narrow subdiaphragmatic area were not considered significant during the abdominal distension. Moreover, a lower insufflation pressure was used for all patients undergoing laparoscopic hiatal hernia repair, further reducing the differences between trocar positioning before and after abdominal insufflation [16,17,18].

Precise reconstruction of the patient’s 3D anatomy from high-resolution medical imaging is mandatory for surgical planning [19]. In our model, we used a variable number of CT scans and mathematical algorithms for interpretation and reconstruction of an accurate anatomical model. The software also calculated the optimal position of the instruments with regard to anatomical landmarks and instrument’s angle.

## 5. Conclusions

Our software offers a very intuitive and useful tool for simulating the position of the trocar’s entry ports for an efficient reach, superior maneuverability, and good exposure for the surgeons, which are essential for reducing the surgical team fatigue, decreasing operating time, and the overall success of the operation. It also proved to be an effective and user-friendly preoperative simulation software for trocar placement, showing its feasibility and safety during the early-stage clinical implementation. Future studies will assess the clinical benefits of the planning system over the conventional, empiric trocar positioning in more patients with other challenging pathologies.

## Figures and Tables

**Figure 1 diagnostics-10-00621-f001:**
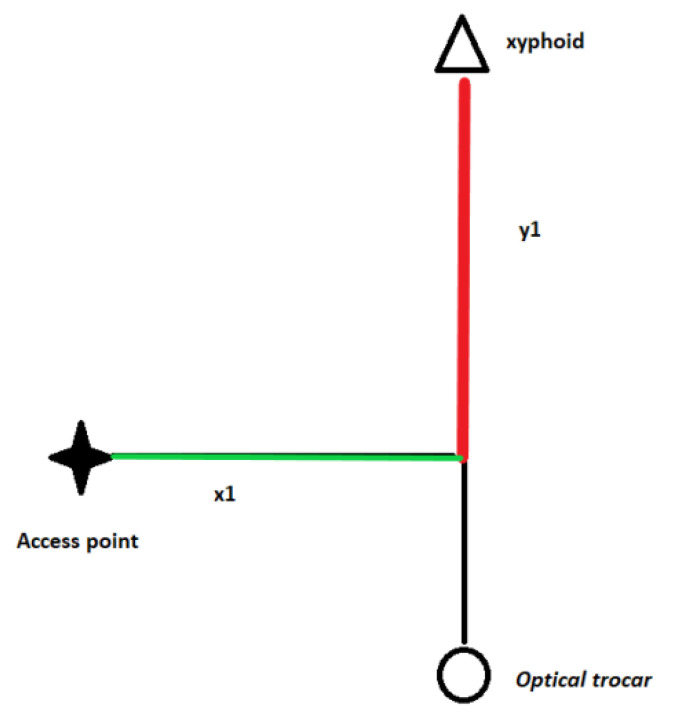
The set of coordinates for the selection of the access point of the trocars. The optical trocar was placed in all three cases on the median (xipho-umbilical) line. x1—distance between the access point and its projection on the xipho-umbilical line; y1—distance between the xiphoid and the projection of the access point on the xipho-umbilical line.

**Figure 2 diagnostics-10-00621-f002:**
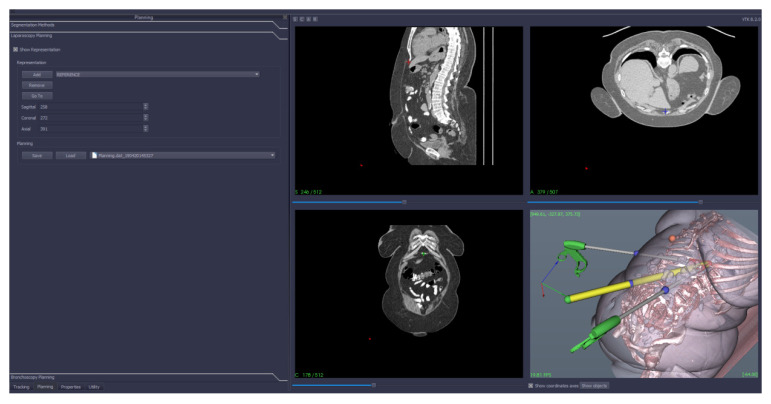
CT-based 3D reconstruction of the abdomen with preoperative instrument positioning simulation.

**Figure 3 diagnostics-10-00621-f003:**
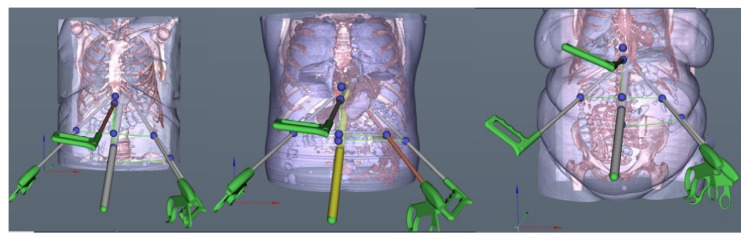
Final aspect of the volumetric reconstruction and laparoscopy planning of the 3 patients using iMTECH software.

**Table 1 diagnostics-10-00621-t001:** The system of coordinates for the access points of the working trocars [P(x_n_y_n_)] and for the optical trocar [P(opt)].

	α_a_	α_m_	P(x_1_y_1_) (mm)	P(x_2_y_2_) (mm)	P(x_3_y_3_) (mm)	P(x_4_y_4_) (mm)	P(opt) (mm)
Patient 1	30°	60°	106.9/128	112.7/128	170.6/192.8	20/0	159.6
Patient 2			120/121.2	113.1/121.2	180.4/141.1	36/0	155.7
Patient 3			114/107.3	113.5/107.3	185.7/182.9	25.7/0	143.1
Patient 4			108.3/117.1	111.6/109.4	177.4/165.6	23/0	158.3
Patient 5			117.3/120.9	109.2/110.3	179.7/150.4	21.2/0	149.1

**Table 2 diagnostics-10-00621-t002:** Clinical and perioperative characteristics of the selected patients.

Patient	Age (yrs.)	Sex	BMI	Hernia Largest Dimension (Cm)	Operative Time (Min.)	Dissection Time (Min.)	Cruroraphy Time (Min.)	Fundoplication Time (Min.)	Postop. Discharge Day (POD)
Patient 1	65	F	37.6	10	200	47	23	31	6th p.o. day
Patient 2	46	M	24.5	8	210	49	25	35	3rd p.o. day
Patient 3	62	F	36.5	6	180	36	20	29	5th p.o. day
Patient 4	50	M	25.3	8	140	44	19	26	4th p.o. day
Patient 5	59	F	28.9	7	120	42	17	25	2nd p.o. day
Control 1	66	F	26.9	9	210	51	29	31	5th p.o. day
Control 2	68	F	29.1	7	180	45	30	30	6th p.o. day
Control 3	82	F	27.3	8	150	44	27	33	3rd p.o. day
Control 4	63	F	28.9	6	180	43	20	35	5th p.o. day
Control 5	68	F	31.3	7	200	53	32	37	4th p.o. day
Control 6	72	F	35.9	8	180	44	23	40	6th p.o. day
Control 7	69	F	31.2	8	190	49	29	31	5th p.o. day
Control 8	74	F	35.1	9	210	55	33	39	4th p.o. day
Control 9	41	F	29.6	10	130	54	21	31	4th p.o. day
Control 10	60	F	28.4	9	215	47	31	41	3rd p.o. day

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
