# Peer review of "Preoperative Computer-Assisted Laparoscopy Planning for the Minimally Invasive Surgical Repair of Hiatal Hernia"

_diagnostics, 2020, doi:10.3390/diagnostics10090621_

Round 1
Reviewer 1 Report
Thank you for allowing me to review this article.
This is very interesting study about preoperative planning of laparoscopy using digital model. I think this approach can support laparoscopic surgeons in their operative activity and this paper adds important innovation in the literature.
The aim of the paper is to show the usefulness of a novel surgical planning and navigation software to determine the optimal location of skin incisions and surgical instrument placement depth in laparoscopy. IN particular the authors tested the application in 3 cases of laparoscopic repair of hiatal hernia.
The main contributions of the article is the novel approach to a common operation in laparoscopic centers, where planning is paramount to establish the correct surgical treatment, especially in such patients with difficult abdomens, like obese patients.
The main findings of this study allowed to test a software to elaborate a 3D reconstruction of the patient on the basis of preoperative CT scan. This model allows to study the anatomical structures and to plan the correct placement of laparoscopic trocars before the operation. So during the operation the surgeons do not have any difficulty during surgical maneuvers, because all the distances between the trocars and the placement of surgical instruments are calculated preoperatively.
The general feedback of the surgeons was that this software significantly improves the ergonomics, was easy to use and particularly useful in obese patients with large hiatal defects.
The main strenght is that this software offers a very intuitive and useful tool for preoperative simulation of trocars position during laparoscopic surgery and I think it should be implemented with further studies.
As minor issue I think that only 3 patients is a small number to demonstrate the usefulness of the software and this is not statistically relevant.
Author Response
Dear Sir,
Thank you for your extensive review. We addressed the issues raised in your review by adding more cases (two patients operated using the iMTECH simulation software during the last 7 months). We also added a control group consisting of 10 patients which underwent floppy Nissen procedures for similar cases of large hiatal hernias in the given interval by the same operating team without the use of iMTECH. However, our main objective was to develop an intuitive, flexible, and easy to use software solution, for preoperative surgical planning and to verify the feasibility of this software on a small number of cases, in order to identify any possible drawbacks and improve its functions.
For any other information we are at your disposal.
Yours,
The authors
Reviewer 2 Report
This manuscript is about utility of a software used for preoperative simulation of laparoscopic surgery for hiatal hernia. This software seems to be well.
I have some questions.
(1) Was the operative time shorter than that of the conventional approach ?
(2) Snce the evaluation method is vague, it is better to carry out a more specific evaluation.
(3) Even if it is a small area, the influence of pneumoperitoneum is likely to be considerable. How was the difference from the simulation? It is necessary to explain the difference.
(4) The number of cases is too small, so it is better to consider a few more cases.
Author Response
Dear Sir,
Thank you for your review. Please allow us to kindly respond to each question that you have asked:
(1) Was the operative time shorter than that of the conventional approach?
This issue was addressed by including a control group in our analysis, consisting of patients undergoing hiatal hernia repair without use of the preoperative simulation software. The results indicated a reduction in the operative time in preoperative simulation group compared to the conventional approach.
(2) Since the evaluation method is vague, it is better to carry out a more specific evaluation.
The first method of evaluation consisted of an objective assessment of relevant clinical and operative parameters (BMI, size of hernia, operative time etc). For a more objective assessment of the feasibility and efficiency, we provided a more extensive analysis of preoperative and intraoperative variables that may have influenced the outcome of each procedure. We also included a control group, as stated above.
The second method of evaluation was based on the feed-back provided by the operative team, which now includes an additional member (involved in the two surgical procedures added). This method is also described in a slightly more detailed manner . The assessment was performed pre- and postoperatively by every member of our operative team, based on a Likert-scale questionnaire. The questionnaire will be provided as a supplementary file to the main manuscript, if the Editors and the reviewers consider it useful.
(3) Even if it is a small area, the influence of pneumoperitoneum is likely to be considerable. How was the difference from the simulation? It is necessary to explain the difference. (Reviewer 2)
We believe it is beyond doubt that the optimal solution for the final “map” of trocar insertion points would be to include as many variables as possible in this simulation software, and this is the final goal of our endeavor, which will hopefully be attained after further improvements, thus making any heuristic approach to trocar insertion obsolete. In this regard, pneumoperitoneum is a distinctively important aspect for every minimally invasive procedure, the distension of the abdominal wall and the changes in its form (geometry) are depending on many factors (elasticity, density of its aponeurotic, muscular structures, degree of adiposity, degree of curarisation, position of the operating table in lateral tilt or Trendelenburg/anti-Trendelenburg position). In our first case that underwent surgery using this planning software we also measured the set of coordinates for each trocar insertion point after the pneumoperitoneum was insufflated and the patient positioned in a 30 degrees procubitus. The variances in position were less than 1 cm and we considered them to be little relevant. We used 9-11 mm Hg insufflation pressure. The differences could be explained by the fact that ribs cage have limited distensibility and the chest CT used for evaluation is performed in deep inspiration, which modifies the rib cage in a way similar to that of the abdominal insufflation. However, on a more general level we acknowledge the need of further improvements in the software capabilities.
(4) The number of cases is too small, so it is better to consider a few more cases.
Additional cases were included (two patients operated using the iMTECH simulation software during the last 7 months). We also added a control group consisting of 10 patients which underwent floppy Nissen procedures for similar cases of large hiatal hernias in the given interval by the same operating team without the use of iMTECH. However, our main objective was to develop an intuitive, flexible, and easy to use software solution, for preoperative surgical planning and to verify the feasibility of this software on a small number of cases, in order to identify any possible drawbacks and improve its functions.
For additional information we are at your disposal.
The authors